# Oxidation of Aqueous Naproxen Using Gas-Phase Pulsed Corona Discharge: Impact of Operation Parameters

Romana Kopecká, Liina Onga and Sergei Preis *

Department of Materials and Environmental Technology, Tallinn University of Technology, 5 Ehitajate Tee, 19086 Tallinn, Estonia
* Correspondence: sergei.preis@taltech.ee; Tel.: +372-6203365

**Abstract:** Naproxen is a widely used non-steroidal anti-inflammatory drug poorly metabolized in the human body, thus resulting in its presence in domestic wastewaters. It is resistant to conventional wastewater treatment, making new methods necessary. Pulsed corona discharge, an energy-efficient advanced oxidation process, was experimentally studied for the oxidation of naproxen in various operation conditions, showing high energy efficiencies in a wide span of pH levels, concentrations, and pulse repetition frequencies. Surfactants present in treated solutions appeared to enhance the degradation rate. The research results contribute to the knowledge of the method's chemistry and technology, supporting its full-scale implementation.

**Keywords:** apronax; electric discharge; hydroxyl radical; plasma; wastewater treatment

## 1. Introduction

The rapid development and consumption of new medicines results in their appearance in the environment, affecting natural conditions. Non-steroidal anti-inflammatory drugs (NSAIDs), known as painkillers alleviating fever and inflammatory processes, represent a wide class of massively used medicines. An example of a non-selective NSAID is naproxen (NPX), a drug used to relieve pain, tenderness, swelling, and fever [1] (Figure 1).

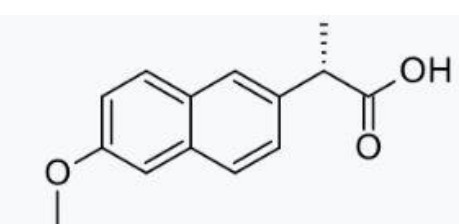

**Figure 1.** Molecular structure of naproxen.

Since its first application in 1976, it has been released into the environment due to its wide use and poor metabolism in the human body [2]. Up to 72% of excreted NPX is present in either original or conjugated form [3]. In the human body, NPX is metabolized into two key products: into 6-O-desmethylnaproxen (DMN) by O-dealkylation and into NPX acyl glucuronide by acyl glucuronidation. The latter metabolite has not been detected neither in human nor rat urine-excreted NPX [4]. The concentrations of NPX in the wastewater treatment plant effluent ranged from 25 ng $L^{-1}$ to 33.9 µg $L^{-1}$ [5]; the major metabolite DMN was quantified at concentrations up to 0.56 µg $L-1$ [6]. Naproxen was detected in the surface water, groundwater, and eventually in drinking water as well [7]. Concentration of NPX detected in fish bile can be up to 1000 times higher than the concentration detected in the lake the fish live in [8]. In case of aquatic organisms, NPX can affect mRNA expression [9]. In humans, NPX can cause gastrointestinal damage

by its non-selective inhibition of cyclooxygenase enzymes, resulting in inflammation and ulceration, and renal system can be affected as well [9,10].

Conventional wastewater processing at treatment plants are able to reduce the NPX content from its almost complete removal to only a 40% degradation level, with the rest ending up in a water recipient [11]. A group of potentially applicable technologies might be advanced oxidation processes (AOPs), the methods where organic compounds, mostly non-biodegradable refractory ones, are being oxidized. The history of AOPs starts in the 1980s when widely defined in 1987 by Glaze et al. [12]. First, they were proposed for drinking water treatment; later, they started to be used for wastewater treatment as well [13]. Hydridooxygen (trivial name 'hydroxyl') radicals [14] provide the advanced character of AOPs and present the distinctive feature of AOPs: HO• radicals are the most powerful non-selective oxidants in water treatment at ambient conditions. Short-living hydridooxygen radicals react with organic compounds as electrophiles, adding to C=C bonds or abstract H from C-H bonds [13].

A representative of AOPs that is potentially applicable for a broad scale of compounds is pulsed corona discharge (PCD), with its high energy efficiency [15–17]. Pulsed corona discharge reactors provide the contact between the gas-phase plasma and treated water showering in droplets, films, and jets through the discharge zone between high-voltage and grounded electrodes [17]. Among other factors, hydrophobicity determines the position of the pollutant molecules towards the gas–liquid interface [16]. The gas-liquid contact surface area is crucial for PCD oxidation, resulting from the fact that HO• radicals are formed at the interface [18,19]. Therefore, the oxidation rate may be enhanced if a surfactant, e.g., sodium dodecyl sulphate (SDS), is present and transports the target pollutant molecule to the interface, which depends on the molecular structure of a pollutant [20].

In this study, aqueous NPX was oxidized using PCD in variable treatment conditions: pH, initial concentration of the NPX solution, pulse repetition frequency, and addition of SDS. The comparison was based on energy efficiencies and the oxidation end products were identified.

## 2. Materials and Methods

Naproxen $\geq$ 98% was purchased from Sigma-Aldrich (USA) and SDS from Lach-Ner, s.r.o. (Czech Republic). All chemicals were of analytical grade used without further purification. All solutions were prepared using bidistilled water with the resistance exceeding 18.2 M$\Omega$ cm$^{-1}$.

The aqueous NPX solution was prepared using NPX powder, bidistilled water, and ten drops of 1-M NaOH per 1.0 L for a better solubility of the compound. Analogously, an SDS solution of various concentrations (100 and 200 mg L$^{-1}$) was prepared.

The pulsed corona discharge device used for the experiments consisted of four main parts—a PCD reactor with an electrode system and perforated plate for water distribution, a storage tank of 40 L, a pulse generator, and a water circulation pump (Figure 2). The reactor contains two vertical parallel plates with horizontally placed string electrodes of 20 m of total length and 0.55 mm in diameter in between. The wire electrodes and grounded plates are distanced 18 mm from each other, so that the horizontal size of the plasma zone measured 36 $\times$ 500 mm. The flow rate of the treated solution was 15.8 L min$^{-1}$ when pumped from the storage tank to the PCD reactor. The experimental solution was dispersed using a perforated plate with 51 perforations of the diameter 1.0 mm. The pulse generator provides high-voltage pulses at frequencies from 50 to 880 pps with respective output power of 9.0 to 123.3 W. The pH of the experimental solution was adjusted at the starting point of the treatment using 5 M aqueous solutions of NaOH or H$_2$SO$_4$.

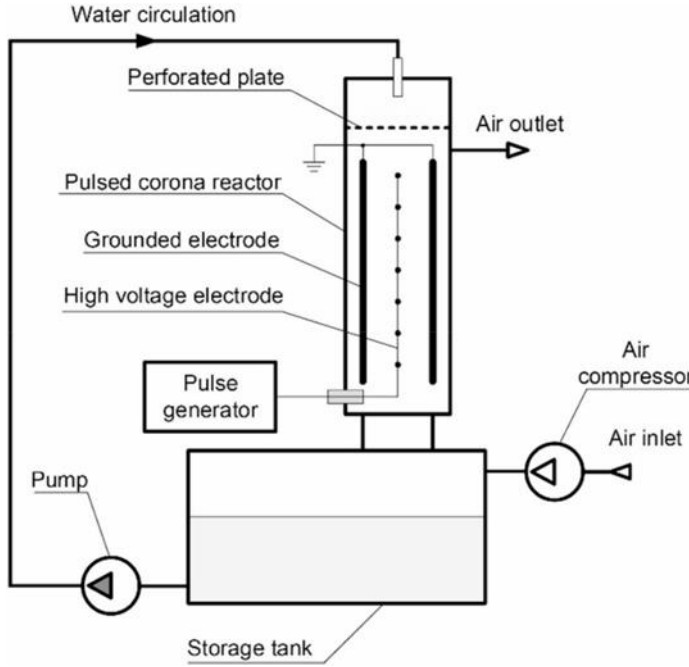

**Figure 2.** PCD experimental device outline.

The pulse generator was switched on for a required time interval, providing a predetermined energy dose delivered to the reactor. The pulse generator was switched off and the NPX solution was mixed in the reactor for three minutes before sampling. The total treatment time varied according to the experiment parameters comprising either 60 or 90 min. All experiments were duplicated with the result deviations fitting into a 5% confidence interval.

The energy efficiency of aqueous NPX oxidation $E$, g kW$^{-1}$h$^{-1}$, was calculated for 90% degradation using Equation (1):

$$E = \frac{\Delta c \cdot V}{\Delta t \cdot P},$$

(1)

where $\Delta c$ is the decrease in NPX concentration, g L$^{-1}$; $V$ is volume of the treated solution, L; $\Delta t$ is time for 90% degradation of NPX, h; $P$ is output power of pulse generator, kW.

Samples were analyzed for NPX content by means of high performance liquid chromatography (HPLC), Shimadzu (Japan), equipped with a Phenomenex Gemini column (150×2.0 mm, 1.7 mm) filled with a stationary phase NX-C18 (110 Å, 5 µm). An isocratic method was used. A mobile phase was composed of 60% of eluate A (water containing 0.3% of formic acid, flow rate 0.1200 mL min$^{-1}$) and 40% of eluate B (acetonitrile containing 0.3% of formic acid, flow rate 0.0800 mL min$^{-1}$). Total flow rate was 0.2 mL min$^{-1}$, injection volume was 100 µL, and oven temperature was 40 °C. The UV-detector GENESYS™ 10S UV-Vis Spectrophotometer by Thermo Scientific™ (USA) was used at the wavelength of 232 nm. The limits of quantification and detection comprised 0.5 and 0.05 mg L$^{-1}$, respectively. pH was measured using a SevenCompact pH meter S220 by Mettler-Toledo Vietnam LLC (Vietnam). Total organic carbon was measured using a multi N/C 3100 TOC analyzer by Analytik Jena GmbH (Germany). Ion chromatography was performed in quantification of end-product anions with chemical suppression of the eluent conductivity, using 761 Compact IC by Metrohm (Switzerland). The device was equipped with METROSEP A Supp 5 column (150 mm × 4.0 mm). The sample injection volume was 20 µL. Separation of ions was attained using an eluent mixture of 3.2 mM Na$_2$CO$_3$ + 1.0 mM NaHCO$_3$ with a flow rate of 0.7 mL min$^{-1}$.

## 3. Results and Discussion

### 3.1. Impact of Initial pH and Concentration of NPX

Experiments were conducted with three initial pH values—acidic pH ~ 3, neutral pH ~ 7, and alkaline pH ~ 11. Later, the pH was not maintained and generally decreased due to the nitric acid formation in the air PCD [21]. The decreased pH is also supported by the formation of carboxylic acids. As a result, the final pH decreased by about two units in alkaline and neutral solutions but remained intact in the acidic range.

The results showed rather effective PCD oxidation of NPX. For example, the pollutant vanished entirely from the solution after less than 16.5 min of treatment at the lowest pulse repetition frequency. pH appeared to moderately influence the oxidation rate in a wide range of the parameter, showing slightly faster oxidation in acidic and circumneutral media. In 5 min of treatment, providing 0.15 kWh m$^{-3}$ of delivered energy, the NPX residual concentration was about 7% and 17.6% of the initial concentration of 10 mg L$^{-1}$ in acidic and alkaline media, respectively (Figure 3a). A somewhat decreased oxidation rate in alkaline solutions may be explained by decreased hydrophobicity of the dissociated NPX molecule, as earlier established for other medical substances [16]. One can see the high efficiency of oxidation, expectedly growing with the target pollutant concentration: a doubled-starting NPX concentration resulted in the oxidation efficiency improved for about 30% (Figure 3b). The energy efficiency of NPX oxidation exceeded 50 g kW$^{-1}$ h$^{-1}$ at the initial concentration of 10 mg L$^{-1}$, which is close to what was observed in rapid phenol oxidation experiments, reaching 55 to 88 g kW$^{-1}$ h$^{-1}$ in air at the initial concentration of phenol at 100 mg L$^{-1}$ [22].

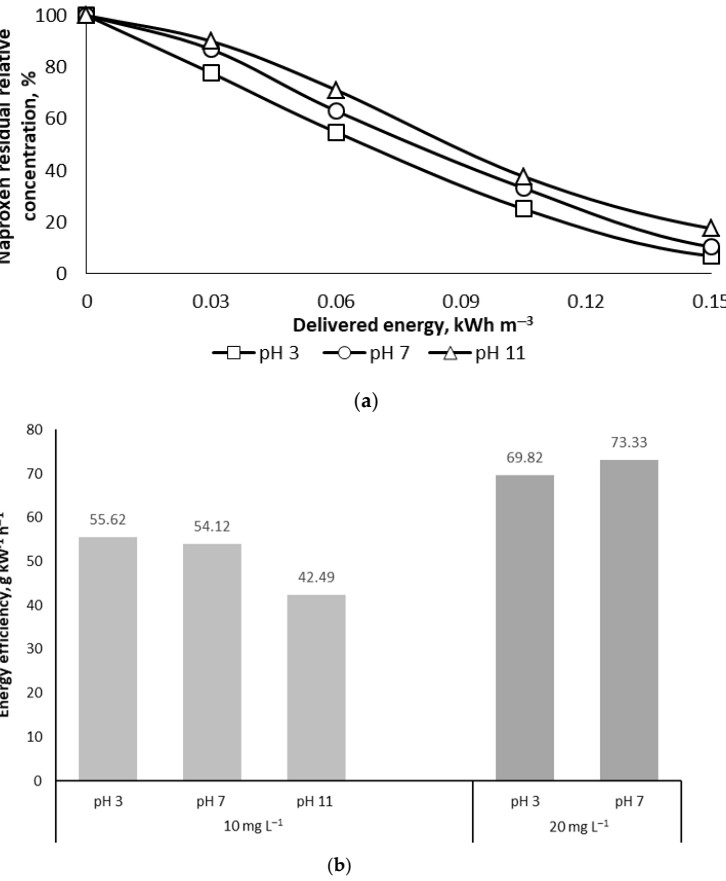

(**a**)

(**b**)

**Figure 3.** Naproxen PCD oxidation: (**a**) residual relative concentration of NPX dependent on delivered pulsed energy at pH$_0$ variations (C$_0$ = 10 mg L$^{-1}$); (**b**) energy efficiency of NPX oxidation, NPX initial concentration 10 or 20 mg L$^{-1}$ at treatment times 60 and 90 min with maximum delivered energy 1.8 and 2.7 kWh m$^{-3}$, respectively; pulse repetition frequency 50 pps (9 W), treated sample volume 5.0 L; oxidation efficiency calculated at 90% of NPX removal.

### 3.2. Impact of Pulse Repetition Frequency

Degradation of NPX at the frequency of 200 pps was more than three times faster than that at 50 pps but when the delivered energy was compared, a similar course of the oxidation was observed. An amount of 93% of NPX was oxidized in both experiments with the delivered energy of 0.167 kWh m$^{-3}$. Only a small difference between the two experimental settings was observed when it comes to energy efficiency; the difference was only 2.5 g kW$^{-1}$ h$^{-1}$ in favor of the lower pulse repetition frequency (Figure 4).

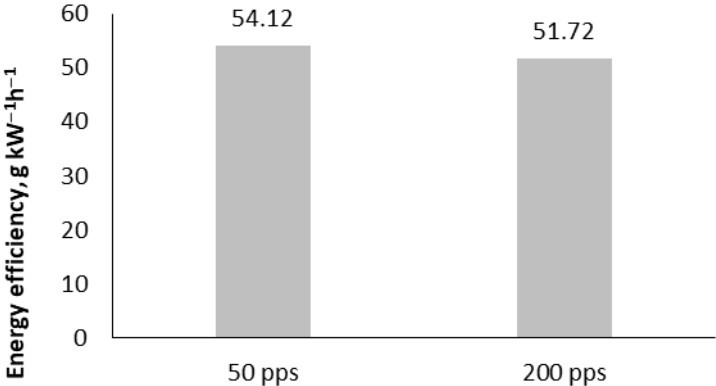

**Figure 4.** Energy efficiency of naproxen oxidation in PCD-treated solutions at pulse repetition frequency variations: NPX initial concentration 10 mg L$^{-1}$; delivered energy 1.8 kWh m$^{-3}$; pH$_0$ ~ 7; treated sample volume 5.0 L; oxidation efficiency calculated at 90% of NPX removal.

The treatment time is longer at lower frequencies, which results from the power being nearly proportional to the pulse repetition frequency. Lower frequencies provide ozone with more time to react between pulses contributing to the degradation. Naproxen is a rapidly reacting compound that might benefit from a higher pulse repetition frequency at shorter treatment times by utilizing all oxidants provided by PCD. The rapidly reacting substances tend to be moderately affected by pulse repetition frequency as previously observed [23] and as observed in this study.

### 3.3. Impact of Sodium Dodecyl Sulphate Addition

The NPX degradation rate noticeably increased with the addition of SDS, showing a minor improvement with a further increase in the concentration of the surfactant. The NPX oxidation efficiency increased from 54.12 to 69.85 g kW$^{-1}$h$^{-1}$, i.e., for about 29%, when 100 mg L$^{-1}$ of SDS was added, whereas a twofold increase in SDS concentration resulted in a negligible change in oxidation efficiency of NPX (Figure 5). The latter is explained by the excess presence of SDS. As established earlier [20], the role of a surfactant promoting PCD oxidation is determined by the interaction of radicalized SDS molecules with the aromatic structure of the target pollutant molecule, thus delivering the NPX molecule to the gas–liquid interface for more effective surface oxidation, with HO• radicals formed at the gas–liquid interface.

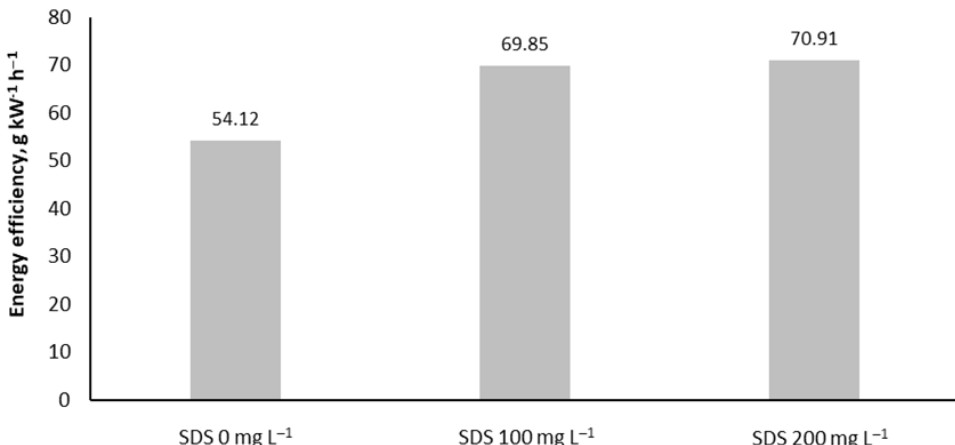

**Figure 5.** Energy efficiency of NPX PCD oxidation dependent on SDS additions: NPX initial concentration 10 mg L$^{-1}$; pulse repetition frequency 50 pps (9 W); maximum delivered energy 1.8 kWh m$^{-3}$; pH$_0$ ~ 7; treated sample volume 5.0 L; oxidation efficiency calculated at 90% of NPX removal.

### 3.4. End Products of Oxidation

Previous studies with NPX oxidation, with Fenton reagent in combination with persulfate, reported transformation products, including 2-methoxy-6-vinylnaphthalene, 2-acetyl-6-methoxynaphthalene, 1-(6-methoxynaphthalen-2-yl)ethyl hydroperoxide, and 2-(6-hydroxy-2-naphthyl)propanoic acid associated with the hydridooxygen radical attack on the side chain of NPX [24]. Further oxidation of intermediates usually results in aromatic ring cleavage and the formation of simple carboxylic acids sought in this study.

The analysis of the end products was conducted for experiments performed at pH ~ 3 and 7 with an NPX starting concentration of about 12.3 mg L$^{-1}$. The results show acetate as the main identified product at its highest concentration of 24.1 mg L$^{-1}$ at the delivered energy dose of 1.8 kWh m$^{-3}$ with a further gradual decrease (Figure 6a). Nitrate expectedly showed a rising tendency in its concentration during the whole experiment as a product of nitric oxide formation in PCD [21]. Formate was identified at low concentrations not exceeding 1.4 mg L$^{-1}$ at 1.8 kWh m$^{-3}$. Oxalate was not quantified due to its peak overlapping sulphate used as sulphuric acid for the pH adjustment in experiments in acidic media [25]. The experiment at the neutral pH$_0$ with no sulphate added was undertaken to avoid the interference in oxalate identification.

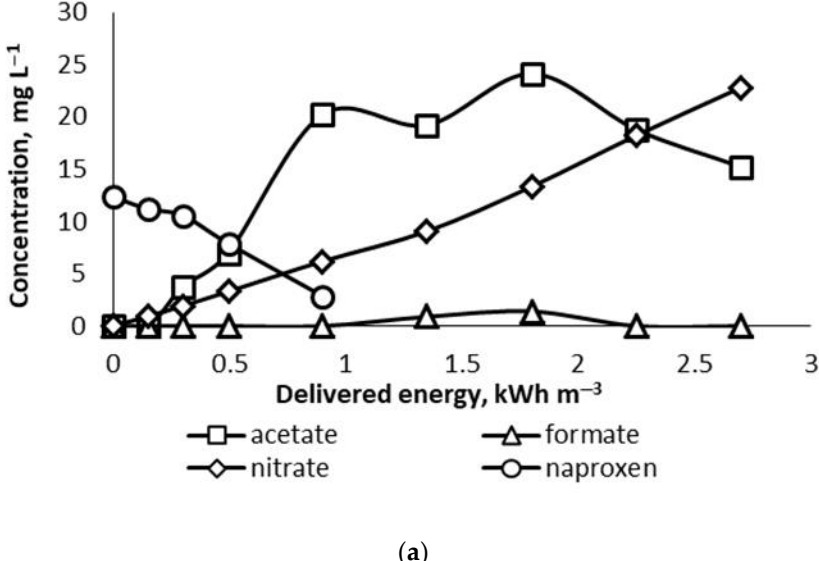

(**a**)

**Figure 6.** *Cont*.

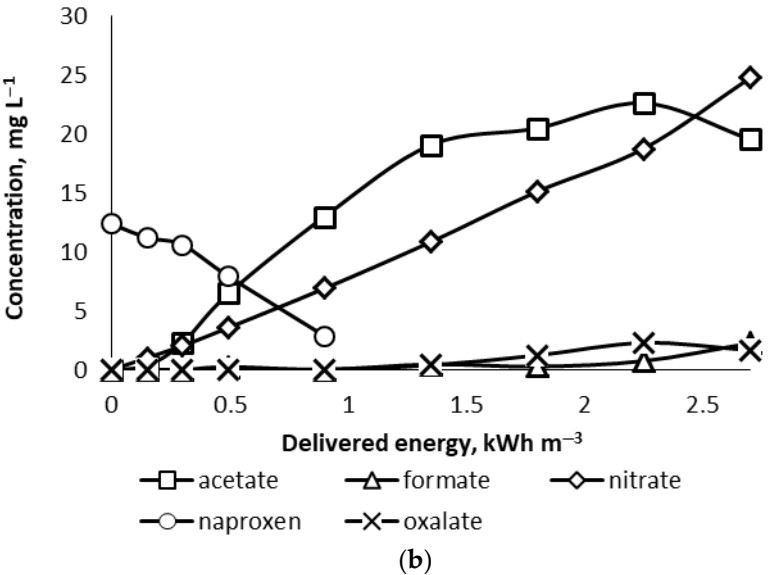

(**b**)

**Figure 6.** Concentrations of end products of PCD oxidation of NPX (**a**) at $pH_0 \sim 3.0$ and (**b**) $pH_0 \sim 7.0$: pulse repetition frequency 50 pps (9 W), volume of treated sample 5.0 L.

At $pH_0 = 7$, the concentrations of identified anions approximately reached the same levels as in acidic conditions (Figure 6b). Since sulphuric acid was not used in the neutral media, oxalate was observed at its maximum concentration of 2.27 mg $L^{-1}$ at the delivered energy of 2.25 kWh $m^{-3}$. This also makes oxalate expectedly formed in acidic PCD-treated NPX solutions, although its concentrations may be compromised by faster oxidation of oxalate in acidic solutions [25]. In contrast, accumulation of acetate appears due to its low oxidation rate. The latter circumstance also explains the fact that the TOC analysis showed minor changes within the experimental conditions. Table 1 presents the carbon balance for the samples treated with energy doses of 0.9, 1.8, and 2.7 kWh $m^{-3}$. The refractory character of NPX oxidation products was also reported by Rosal et al. [26].

**Table 1.** Total organic carbon balance of NPX oxidation in PCD.

| pH | 3.0 | | | | 7.0 | | | |
|---|---|---|---|---|---|---|---|---|
| Delivered energy dose, kWh $m^{-3}$ | 0 | 0.9 | 1.8 | 2.7 | 0 | 0.9 | 1.8 | 2.7 |
| | | | Concentration, mg $L^{-1}$ | | | | | |
| NPX | 12.3 (8.98) | 1.80 (1.30) | 0.0 | 0.0 | 12.3 (8.98) | 3.15 (2.30) | 0.0 | 0.0 |
| Acetate | 0.0 | 19.1 (7.64) | 21.9 (8.76) | 18.2 (7.28) | 0.0 | 13.5 (5.40) | 21.4 (8.56) | 19.9 (7.96) |
| Oxalate | 0.0 | - | - | - | 0.0 | 0.0 | 1.15 (0.31) | 2.01 (0.54) |
| Formate | 0.0 | 0.0 | 1.4 (0.37) | 0.0 | 0.0 | 0.0 | 0.5 (0.13) | 2.04 (0.53) |
| Sum TOC, mg C $L^{-1}$ | 8.98 | 8.94 | 9.13 | 7.28 | 8.98 | 7.70 | 9.00 | 9.03 |
| Measured TOC, mg C $L^{-1}$ | 9.00 | 8.99 | 9.04 | 7.31 | 9.00 | 8.93 | 9.01 | 8.99 |

Note(s): TOC values are given in parentheses.

Regarding the molar concentration of NPX in treated solutions, the amount of carbon-containing oxidation products, mostly acetate, may be estimated in their theoretically expected and real amounts. The molar concentration of NPX with a molar mass of 230 g $mol^{-1}$ in the solution containing 12.3 mg $L^{-1}$ comprises 0.053 mmol $L^{-1}$. The maximum concentration of acetate observed in the experiments comprised 24.1 mg $L^{-1}$, which corresponds to a molar concentration of 0.4 mmol $L^{-1}$. This latter amount is close to the theoretically possible formation of acetate in the stoichiometric amount of 7:1 with NPX, containing fourteen carbon atoms, which exhibits the difference in mass concentrations of NPX and acetate due to an increased oxygen-to-carbon ratio compared to what is expected for oxidation.

## 4. Conclusions

This study showed a rapid oxidation of NPX by PCD with high efficiency increasing with the increased concentration.

The study showed that pulse repetition frequency has only a small impact on the NPX degradation rate explained by the entirely consumed reactive oxidant species, irrespective of the time between the discharge pulses. Therefore, time can be saved by the use of high repetition frequencies for a higher intensity process at the same efficiency numbers.

The initial pH does not seem to be a crucial factor affecting the oxidation; however, acidic conditions happen to be somewhat beneficial for NPX degradation.

The presence of SDS surfactant enhances the NPX degradation rate to a noticeable extent, which is explained by the interaction of NPX molecules with the SDS radicals, providing the target molecules with oxidation in closer proximity to the plasma–liquid interface.

Identification of oxidation products performed using ionic chromatography showed carboxylic anions, with acetate mainly being the non-harmful ultimate oxidation product in amounts close to what is stoichiometrically expected.

**Author Contributions:** Conceptualization, L.O. and S.P.; formal analysis, R.K. and L.O.; funding acquisition, S.P.; investigation, R.K.; methodology, L.O. and S.P.; project administration, S.P.; resources, S.P.; supervision, L.O. and S.P.; validation, R.K., L.O., and S.P.; visualization, R.K.; writing—original draft, R.K.; writing—review and editing, S.P. All authors have read and agreed to the published version of the manuscript.

**Funding:** This research was funded by the Institutional Development Program of Tallinn University of Technology for 2016–2022, project 2014–2020.4.01.16-0032, from the EU Regional Development Fund.

**Institutional Review Board Statement:** Not applicable.

**Informed Consent Statement:** Not applicable.

**Data Availability Statement:** Not applicable.

**Conflicts of Interest:** The authors declare no conflict of interest.

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
