# Peer review of "Oxidation of Aqueous Naproxen Using Gas-Phase Pulsed Corona Discharge: Impact of Operation Parameters"

_water, doi:10.3390/w14203327_

Round 1
Reviewer 1 Report
(1) The writing style of radicals should be corrected according to the following reference: “Koppenol, W. (2000). "Names for inorganic radicals (IUPAC Recommendations 2000)." Pure and Applied Chemistry 72(3): 437-446.”
(2) Some relevant references, such as Chemical Engineering Journal 409: 128176, Separation and Purification Technology 288: 120716, Chemical Engineering Journal 451: 138588, Chinese Chemical Letters 33(4): 2125-2128 should be citied in introduction.
(3) The sequence number of “The PCD experimental device outline” should be Figure 2, pls modified.
(4) Add the reference in Line 98.
(5) There are only two or three levels for investigated operation parameters, maybe 4-5 is more convincing.
(6) More recently references should be cited.
Author Response
Reviewer 1
Reviewer’s remark
(1) The writing style of radicals should be corrected according to the following reference: “Koppenol, W. (2000). "Names for inorganic radicals (IUPAC Recommendations 2000)." Pure and Applied Chemistry 72(3): 437-446.”
Authors’ response
The authors agree with the systematic approach presented in the referred source and provide the systematic name of hydroxyl-radical ‘hydridooxygen’ in the list of key words.
Reviewer’s remark
(2) Some relevant references, such as Chemical Engineering Journal 409: 128176, Separation and Purification Technology 288: 120716, Chemical Engineering Journal 451: 138588, Chinese Chemical Letters 33(4): 2125-2128 should be citied in introduction.
Authors’ response
The authors thank the reviewer for attracting attention to the publications of the Professor He Huan’s group from Nanjing Normal University, dealing with peroxo-monosulphate application in catalytic oxidation of medical substances, particularly, sulfamethoxazole. The authors, however, worked on application of pulsed corona discharge (PCD) towards naproxen oxidation, without addition of extrinsic persulphates. The combination of PCD with peroxide and persulphate extrinsic oxidants was studied by the authors earlier; the results were published in the following papers:
Tikker, P., Dulova, N., Kornev, I., Preis, S. (2021) Effects of persulfate and hydrogen peroxide on oxidation of oxalate by pulsed corona discharge, Chem. Eng. J., 411, article number 128586, DOI: 10.1016/j.cej.2021.128586
Onga, L., Kattel-Salusoo, E., Preis, S., Dulova, N. (2022) Degradation of anti-inflammatory drug dexamethasone by pulsed corona discharge: the effect of peroxycompounds addition, J. Environ. Chem. Eng., 10, article number 108042, DOI: 10.1016/j.jece.2022.108042
Nikitin, D., Balpreet Kaur, Preis, S., Dulova, N. (2022) Persulfate contribution to photolytic and pulsed corona discharge oxidation of metformin and tramadol in water, Process Safety and Environmental Protection, 165, 22-30. DOI: 10.1016/j.psep.2022.07.002
The authors experienced difficulties in finding relevance of the suggested citations to the subject of the manuscript: naproxen is of different molecular structure compared to methyl paraben or SMX; PCD provides oxidation mainly with OH-radicals and ozone, not involving persulphate chemistry, which was not studied in this manuscript. The results might be compared in the efficiency of two different methods, as it was made for oxalate, dexamethasone, metformin and tramadol in the authors’ publications, but data on naproxen oxidation with persulphate of PMS are lacking.
Reviewer’s remark
(3) The sequence number of “The PCD experimental device outline” should be Figure 2, pls modified.
Authors’ response
The authors were surprised to see the mistake with the Figures’ numbering in the PDF file. The manuscript downloaded from the MDPI site shows Figures numbered in perfect order.
Reviewer’s remark
(4) Add the reference in Line 98.
Authors’ response
The authors are confused by the changes made by transformation of the MS Word file of the manuscript to the PDF-format made by MDPI submission. In the original manuscript, the following sentence is placed:
The energy efficiency of aqueous NPX oxidation E, g kW-1h-1, was calculated for 90% degradation using Eq. 1:
Reviewer’s remark
(5) There are only two or three levels for investigated operation parameters, maybe 4-5 is more convincing.
Authors’ response
The authors agree with the reviewer, that the more parameters are studied, the better it is. However, the authors studied some other parameters influencing the process efficiency, such as temperature and electric conductivity of the treated solutions, which is considered by the authors as sufficient for the time being. See, for example:
Onga, L., Kornev, I., Preis, S. (2020) Oxidation of reactive azo-dyes with pulsed corona discharge: surface reaction enhancement, J. Electrostatics, 103, article number 103420. DOI: 10.1016/j.elstat.2020.103420
Reviewer’s remark
(6) More recently references should be cited.
Authors’ response
The authors cited sufficient number of supporting references, although relevant references advised by the reviewers may be definitely introduced to the manuscript.
Reviewer 2 Report
The manuscript ‘Oxidation of aqueous naproxen using gas-phase pulsed corona 2 discharge: impact of operation parameters’ deals with the use of a less studied advanced oxidation process, PCD, and its use to oxidize a pharmaceutical model compound.
The paper is well structured, the description of the processes is accurate and the discussion of the results is, in general correct. (Please check errors in lines 82, 98)
There are a couple of facts that cause uncertainty about the obtained results. The solubility of Naproxen is 16 mg/L and has a pKa=4.15 (PubChem data). The authors employed 20 mg/L in some experiments; therefore, it is possible that some of the target pollutant wasn´t completely dissolved. How can this fact affect to the efficiency of the process?
As the authors checked different working pH values and the compound could be present both in the neutral and in the anionic form, ¿Can this fact affect the solubility and the presence of the pollutant in the gas-liquid interface?
Please clarify these important aspects before the publication of the manuscript
Author Response
Reviewer 2
Reviewer’s remark
The manuscript ‘Oxidation of aqueous naproxen using gas-phase pulsed corona 2 discharge: impact of operation parameters’ deals with the use of a less studied advanced oxidation process, PCD, and its use to oxidize a pharmaceutical model compound.
The paper is well structured, the description of the processes is accurate and the discussion of the results is, in general correct. (Please check errors in lines 82, 98)
Authors’ response
The authors are confused by the changes made by transformation of the MS Word file of the manuscript to the PDF-format, made by MDPI submission. In the original manuscript, there are no mistakes.
Reviewer’s remark
There are a couple of facts that cause uncertainty about the obtained results. The solubility of Naproxen is 16 mg/L and has a pKa=4.15 (PubChem data). The authors employed 20 mg/L in some experiments; therefore, it is possible that some of the target pollutant wasn´t completely dissolved. How can this fact affect to the efficiency of the process?
Authors’ response
The authors used and confirmed the following data from Sigma-Aldrich: Naproxen … is lipid soluble, practically insoluble in water at low pH and freely soluble in water at high pH. In previous studies, concentrations of 15 mg/L were used at pH 3 as well [24].
In the manuscript, the authors wrote:
The aqueous NPX solution was prepared using NPX powder, bi-distilled water and ten drops of 1-M NaOH per 1.0 L for a better solubility of the compound.
The authors did not see any obstacle in dissolving naproxen in water. See also the answer to the next remark.
Reviewer’s remark
As the authors checked different working pH values and the compound could be present both in the neutral and in the anionic form, ¿Can this fact affect the solubility and the presence of the pollutant in the gas-liquid interface?
Authors’ response
According to the ALogPS software prediction, the solubility of naproxen in water reaches as much as 29.9 mg/L. The majority of experiments were conducted with the NPX concentration of 10 mg/L thus making no problem with solubility. As to the presence of NPX at the gas-liquid interface, the present research deals with the issue conducting experiments with the surfactant showing a pronounced impact onto the oxidation efficiency.
Reviewer 3 Report
Authors present an interesting work on the degradation of naproxen. Nonetheless, the scientific quality of the manuscript presented falls bellow the publication standard of the Water journal. Please address the following issues prior to submission elsewhere:
1. Writing should be improved. Some sentences in the abstract and introduction need commas to have grammatical sense. Please revise the punctuation throughout the whole manuscript.
2. Also, English should be also revised throughout the text, paying special attention to grammatic errors (such as concordance between subjects and verbs when using singular or plural subjects). Revision by a native speaker is highly recommended.
3. Introduction: authors state that NPX has been also detected in surface water, groundwater and in drinking water. Please specify the measured concentrations.
4. Line 43 - NPX elimination 100% to 40%. Is the 100% data correct? If there is a total NPX elimination there is no point on applying AOPs.
5. Definition for AOPs should be revised. Initially they were based only on hidroxyl radical generation, but other species such as ozone, sulfate radicals, etc. have been employed for this purpose. Please revise the literature to provide a more precise definition on AOPs.ç
6. Revise the equations in the manuscript. Currently eqs are presented with the following text: "Error! Reference source not found"
7. Analytical methods: HPLC, please specify the LOD and LOQ for NPX.
8. Analytical methods: IC, please specify the column and eluent (type of eluent and flow) used in IC analyses.
9. Figure 1. Please specify in the Figure caption the total reaction time.
10. Energy efficiency should be compared to other works for NPX removal.
11. Reaction byproducts. If the initial NPX concentration is 12.3 mg/L, how can you have 25 mg/L nitrate or acetate concentration?? This makes absolutely no sense.
12. Since authors have measured TOC and reaction intermediates, please perform a carbon balance calculating the carbon from NPX and byproducts measured concentrations and comparing it to the measured TOC.
13. Ecotoxicity tests before and after reaction are highly recommended
14. Comparison between the obtained results and other NPX elimination technologies tested (literature) is mandatory to present a good results discussion.
15. TOC results must be presented and discussed.
16. Reactions in ultrapure water using a high NPX concentration (since its found in ppb in WWTP) and adding a surfactant is not really representative. Assays in relevant environmental matrices (river water, WWTP effluent) must be conducted, lowering the NPX initial concentration employed.
17. The role of hydroxyl radicals is not demonstrated. Please perform an experiment in presence of tert-butanol as HO· scavenger to confirm the predominant role of HO·
Author Response
Reviewer 3
Reviewer’s remark
Authors present an interesting work on the degradation of naproxen. Nonetheless, the scientific quality of the manuscript presented falls bellow the publication standard of the Water journal. Please address the following issues prior to submission elsewhere:
Authors’ response
The authors may suggest the categorical opinion of the reviewer derived from a few circumstances described in the manuscript, but missed by the reviewer. The authors tried to bring more clarity to the remarks under the discussion.
Reviewer’s remark
- Writing should be improved. Some sentences in the abstract and introduction need commas to have grammatical sense. Please revise the punctuation throughout the whole manuscript.
Authors’ response
Punctuation checked throughout the manuscript as required.
Reviewer’s remark
- Also, English should be also revised throughout the text, paying special attention to grammatic errors (such as concordance between subjects and verbs when using singular or plural subjects). Revision by a native speaker is highly recommended.
Authors’ response
English was double-checked by the native speaker as required to avoid grammar and syntax errors.
Reviewer’s remark
- Introduction: authors state that NPX has been also detected in surface water, groundwater and in drinking water. Please specify the measured concentrations.
Authors’ response
The authors provide a reference [7] to the review paper presenting a wide spectrum of NPX detection facts in a long list of rivers over the world. At this background, repeating the numbers in the manuscript contributing to the knowledge in NPX oxidation seems either excessive, if all the Table 1 [7] content is copied, or fragmentary, if only the lowest and the highest limits of concentrations are given. The manuscript handles the application of novel technology, bringing sufficient proof for the NPX occurrence in waters.
Reviewer’s remark
- Line 43 - NPX elimination 100% to 40%. Is the 100% data correct? If there is a total NPX elimination there is no point on applying AOPs.
Authors’ response
The statement of the reviewer is arguable in part of having no point in applying AOPs, although the authors must soften the assessment. Marco-Urrea et al. (2010) cited in the manuscript as [11] provide this: the removal of naproxen in wastewater treatment plants is significantly different and ranges from its almost complete removal to only a 40% degradation level. The authors replace 100% formulation to the cautious citation of ‘ranges from its almost complete removal to only a 40% degradation level’.
Reviewer’s remark
- Definition for AOPs should be revised. Initially they were based only on hidroxyl radical generation, but other species such as ozone, sulfate radicals, etc. have been employed for this purpose. Please revise the literature to provide a more precise definition on AOPs.ç
Authors’ response
The authors did not give a definition of AOPs, i.e. there is nothing to revise. In the manuscript, the following text is given, which does not contain erroneous statements:
A group of potentially applicable technologies might be advanced oxidation processes (AOPs), the methods where organic compounds, mostly non-biodegradable refractory ones, are being oxidized. The history of AOPs starts in 1980s when widely defined in 1987 by Glaze et al. [12].
The authors do not have intention to define AOPs on a historical review basis in the manuscript dealing with the technological aspects of PCD application to NPX oxidation. If the definition given by Professor Glaze in the cited article was not historically the first one, this statement may be removed from the manuscript.
Reviewer’s remark
- Revise the equations in the manuscript. Currently eqs are presented with the following text: "Error! Reference source not found"
Authors’ response
The authors are confused by the changes made by transformation of the MS Word file of the manuscript to the PDF-format made by MDPI submission. In the original manuscript, the following sentence is placed:
The energy efficiency of aqueous NPX oxidation E, g kW-1h-1, was calculated for 90% degradation using Eq. 1:
Reviewer’s remark
- Analytical methods: HPLC, please specify the LOD and LOQ for NPX.
Authors’ response
The detection and quantification limits were added as required.
Reviewer’s remark
- Analytical methods: IC, please specify the column and eluent (type of eluent and flow) used in IC analyses.
Authors’ response
The manuscript was amended accordingly.
Reviewer’s remark
- Figure 1. Please specify in the Figure caption the total reaction time.
Authors’ response
The conversion of the submitted MS Word file of the manuscript to the PDF-format mixed up the Figures’ numbers, which, most probably, was reflected in the reviewer’s remark: Figure 1 reflects the NPX structure. The authors may guess that the remark concerns Figure 3, the caption of which looks as follows and contains the required information (underlined):
Figure 3. Naproxen PCD-oxidation: (a) Residual relative concentration of NPX dependent on delivered pulsed energy at pH0 variations (C0=10 mg L-1); (b) Energy efficiency of NPX oxidation: NPX initial concentration 10 or 20 mg L-1 at treatment times 60 and 90 min with maximum delivered energy 1.8 and 2.7 kWh m-3, respectively; pulse repetition frequency 50 pps (9 W), treated sample volume 5.0 L; oxidation efficiency calculated at 90% of NPX removal
The required information was also given in the text.
Reviewer’s remark
- Energy efficiency should be compared to other works for NPX removal.
Authors’ response
The authors would be more than happy to bring the comparison in energy efficiency between methods; this was the standard approach of the authors in all previous publications partly cited in the manuscript [15, 18, 19, 21, 23 and 25]. The efficiency of oxidation, as was observed here, depends largely on the pollutant concentration. The authors spent significant efforts looking for the comparison in literature, but failed to find works handling oxidation of NPX at conditions suitable for reliable comparison or giving sufficient details for calculation of the energy expense. For example, the work of R. Rosal, A. Rodrı´guez., M.S. Gonzalo, and E. Garcı´a-Calvo ‘Catalytic ozonation of naproxen and carbamazepine on titanium dioxide’ published in Applied Catalysis B: Environmental 84 (2008) 48–57 [24], does consider oxidation of NPX at its starting concentration of 15 mg L-1 well suitable for the comparison, but the authors limited their report to the fact that ‘naproxen and carbamazepine were completely consumed in the first few minutes of reaction’, putting all the efforts on the NPX and CMZ mineralization, i.e. TOC removal kinetics at the late stages of treatment within extended time, making their finding irrelevant to the efficiency numbers discussed in the manuscript.
Reviewer’s remark
- Reaction byproducts. If the initial NPX concentration is 12.3 mg/L, how can you have 25 mg/L nitrate or acetate concentration?? This makes absolutely no sense.
Authors’ response
Formation of acetate fits well into the stoichiometry of the reaction: practically all NPX carbon transforms to acetate. The manuscript contains the following explanation right after the Figure 6 missed by the reviewer:
Approaching from the molar concentration of NPX in treated solutions, the amount of carbon-containing oxidation products, mostly acetate, may be estimated in their theoretically expected and real amounts. The molar concentration of NPX with molar mass of 230 g mol-1 in the solution containing 12.3 mg L-1 comprises 0.053 mmol L-1. The maximum concentration of acetate observed in the experiments comprised 24.1 mg L-1, which corresponds to molar concentration of 0.4 mmol L-1, which is close to the theoretically possible formation of acetate in stoichiometric amount 7:1 with NPX, containing fourteen carbon atoms exhibiting the difference in mass concentrations of NPX and acetate due to increased oxygen-to-carbon ratio expected for oxidation.
As a comment, the difference in the mass of oxygenated acetate exceeding the one of NPX makes perfect sense: acetate with the carbon-to-oxygen ratio of 1:1 is substantially heavier than the carbon contained in the NPX molecule with the carbon-to-oxygen atomic ratio 14:3.
As to the nitrate formation, the following explanation was given in the manuscript:
Nitrate expectedly showed a rising tendency in its concentration during the whole experiment as a product of nitric oxides formation in PCD [20].
Explaining that, the authors point to the absence of nitrogen in the NPX molecule; nitric oxides are formed in atmospheric nitrogen oxidation in PCD, the yield of nitrate is irrelevant towards the NPX initial content.
Reviewer’s remark
- Since authors have measured TOC and reaction intermediates, please perform a carbon balance calculating the carbon from NPX and byproducts measured concentrations and comparing it to the measured TOC.
Authors’ response
The authors are grateful to the reviewer, the TOC analysis slipped out of the authors’ attention. This happens because of its negligible changes within the experiments; the authors amended the manuscript as follows:
In contrast, accumulation of acetate appears due to its low oxidation rate. The latter circumstance explains also the fact that the TOC analysis showed negligible changes within the experimental conditions. The refractory character of NPX oxidation products was reported also by Rosal et al. [24].
Reviewer’s remark
- Ecotoxicity tests before and after reaction are highly recommended
Authors’ response
The authors agree that the toxicity of intermediates is important for judging the PCD safety. However, unlike medium-refractory dexamethasone (Onga et al., 2022), the rate of NPX oxidation appeared to be rapid, leaving little chance for potentially more toxic intermediates to increase the water toxicity. As to the end products, the carbon balance given in the manuscript shows the dominant production of acetate, i.e. the authors decided to omit the costly toxicity test.
Reference
Onga, L., Kattel-Salusoo, E., Trapido, M., Preis, S. (2022) Oxidation of aqueous dexamethasone solution by gas-phase pulsed corona discharge, Water, 14, article number 467, https://doi.org/10.3390/w14030467
Reviewer’s remark
- Comparison between the obtained results and other NPX elimination technologies tested (literature) is mandatory to present a good results discussion.
Authors’ response
The reviewer’s remark repeats the one No. 10.
Reviewer’s remark
- TOC results must be presented and discussed.
Authors’ response
The reviewer’s remark repeats the one No. 12.
Reviewer’s remark
- Reactions in ultrapure water using a high NPX concentration (since its found in ppb in WWTP) and adding a surfactant is not really representative. Assays in relevant environmental matrices (river water, WWTP effluent) must be conducted, lowering the NPX initial concentration employed.
Authors’ response
Higher concentrations of the target pollutant provide reliability in results and analyses under the treatment conditions studied in respect of the novel PCD techniques under the scope applied towards the pollutant of emerging concern. At this stage, there is no need to make the study sophisticated and expensive with all the circumstances of the target pollutant micro-concentrations and admixtures competing the target reaction. This is the next step in PCD technology development. Requiring of doing more is a straightforward and simple way of criticism, although the trustworthy character of obtained new knowledge is a subject of review.
Reviewer’s remark
- The role of hydroxyl radicals is not demonstrated. Please perform an experiment in presence of tert-butanol as HO· scavenger to confirm the predominant role of HO·
Authors’ response
The role of radicals and ozone was studied in detail many times in the authors’ previous publications, demonstrating the surface character of the radical attack at the plasma-liquid interface, see, e.g., [15, 18, 19, 21, 23 and 25]. In these articles, the approach suggested by the reviewer was applied with the conclusion of minor role of OH-radical in bulk reactions, whereas the surface oxidation was substantially influenced by the surfactant radical scavengers used in the present research. The manuscript solved other problems, the radical involvement in reactions received sufficient enlightenment earlier. In the manuscript, the authors focused their efforts on the treatment conditions for the rapidly oxidized NPX.
Reviewer 4 Report
The manuscript entitled " Oxidation of aqueous naproxen using gas-phase pulsed corona discharge: impact of operation parameters" has been read carefully. Although the authors have performed a complete evaluation of the oxidation of aqueous naproxen using gas-phase pulsed corona discharge, it is hard to find the significance and advances of this research. Obviously, this article lacks scientific innovations and cannot meet the standard for publishing in Water. Some important issues should not be ignored:
(1) The figures in the paper are not arranged in order
(2) The data in the paper needs to be explained.
(3) Error bars should be added in figures.
Author Response
Reviewer 4
Reviewer’s remark
The manuscript entitled " Oxidation of aqueous naproxen using gas-phase pulsed corona discharge: impact of operation parameters" has been read carefully. Although the authors have performed a complete evaluation of the oxidation of aqueous naproxen using gas-phase pulsed corona discharge, it is hard to find the significance and advances of this research. Obviously, this article lacks scientific innovations and cannot meet the standard for publishing in Water. Some important issues should not be ignored:
Authors’ response
The novelty remained unnoticed by the reviewer – application of novel PCD with unequalled energy efficiency among AOPs known nowadays to oxidation of NPX described for the first time. The interaction with the surfactant improving the oxidation energy efficiency in the interface reaction is also a non-trivial knowledge. The superficial remarks given by the reviewer – numbers of Figures, error bars, - witness a superfluous character of reading. The remark (2) given per se remains unclear: all the manuscript explains the data obtained in the research. Rejection based on the remarks given by the reviewer is baseless and has nothing to do with careful reading.
Reviewer’s remark
(1) The figures in the paper are not arranged in order
Authors’ response
The authors were surprised to see the mistake with the Figures’ numbering in the PDF file. The manuscript downloaded from the MDPI site shows Figures numbered in perfect order.
Reviewer’s remark
(2) The data in the paper needs to be explained.
Authors’ response
The authors are ready to give necessary explanations provided the need in explanations is described by the reviewer.
Reviewer’s remark
(3) Error bars should be added in figures.
Authors’ response
The error bars were not placed to the Figures for the experimental confidence interval given in the text as follows:
All experiments were duplicated with the results deviations fitting into 5-% confidence interval.
Reviewer 5 Report
The theme is interesting but there are some comments that must be addressed by the authors as follows:
1. Line 60-63 Why sodium dodecyl sulphate was used in this study? How about other kinds of surfactant?
2. Please supplement some explanations of the impact of initial pH on naproxen removal.
3. Line 98 there’s an error.
4. Wrong number of some figures.
Author Response
Reviewer 5
Reviewer’s remark
The theme is interesting but there are some comments that must be addressed by the authors as follows:
- Line 60-63 Why sodium dodecyl sulphate was used in this study? How about other kinds of surfactant?
Authors’ response
Non-ionic surfactant IGEPAL C-630 was used in previous works [25] with the scavenging effect on the surface-borne radicals observed for the first time. The cationic SDS surfactant is the one of the vast usage, making it interesting not only from the theoretical standpoint, but from the practical as well. In the SDS impact studies, the authors also disclosed the variable impact of this surfactant on the energy efficiency of oxidation of pollutants [19], i.e. the approach has a preliminary history of studies, referred in the manuscript.
Reviewer’s remark
- Please supplement some explanations of the impact of initial pH on naproxen removal.
Authors’ response
The manuscript was amended as follows:
Somewhat decreased oxidation rate in alkaline solutions may be explained by decreased hydrophobicity of dissociated NPX molecule, as established for other medical substances earlier [15].
Reviewer’s remark
- Line 98 there’s an error.
Authors’ response
The authors are confused by the changes made by transformation of the MS Word file of the manuscript to the PDF-format made by MDPI submission. In the original manuscript, the following sentence is placed:
The energy efficiency of aqueous NPX oxidation E, g kW-1h-1, was calculated for 90% degradation using Eq. 1:
Reviewer’s remark
- Wrong number of some figures.
Authors’ response
The authors were surprised to see the mistake with the Figures’ numbering in the PDF file. The manuscript downloaded from the MDPI site shows Figures numbered in perfect order.
Round 2
Reviewer 1 Report
The writing style of radicals should be corrected according to the following reference: “Koppenol, W. (2000). "Names for inorganic radicals (IUPAC Recommendations 2000)." Pure and Applied Chemistry 72(3): 437-446.”
Author Response
Reviewer 1
Reviewer’s remark
The writing style of radicals should be corrected according to the following reference: “Koppenol, W. (2000). "Names for inorganic radicals (IUPAC Recommendations 2000)." Pure and Applied Chemistry 72(3): 437-446.”
Authors’ response
The authors agree with the systematic approach presented in the referred source and replace ‘hydroxyl radical’ term with ‘hydridooxygen’ throughout the manuscript with the reference to the advised source [14].
Reviewer 2 Report
After the authors' clarification on the solubility of the model compound used, I consider that the manuscript is of sufficient quality to be published.
Author Response
Reviewer 2
Reviewer’s remark
After the authors' clarification on the solubility of the model compound used, I consider that the manuscript is of sufficient quality to be published.
Authors’ response
The authors appreciate the reviewer’s opinion.
Reviewer 3 Report
Authors have tried to correct part of the flaws presented in the original manuscript. Still, some issues must be polished. When reviewers suggest new experiments, these are directed towards improving the quality of the manuscript, not as a simple way of criticism as stated by the authors. Be aware that other people are dedicating time to reading, analyzing and suggesting how your work can be improved, before being bold and disrespectful on your answer.
I suggest authors take into account these suggestions for the present work:
1. Units coherence. All along the manuscript, units are refered as g L-1. In Figure 3, SDS concentration is presented as mg/l. Uniformize the units format and revise throuhout the manuscript.
2. Line 195, revise nomenclature for hydroxyl radical.
3. Authors state that TOC removal is negligible with the following statement: "The latter circumstance explains also the fact that the TOC 214 analysis showed negligible changes within the experimental conditions. The refractory 215 character of NPX oxidation products was reported also by Rosal et al. [24]." In the article by Rosal et al., a 75% TOC reduction is reached. Hence, this does not explain at all your results.
4. The suggestion of performing a carbon balance is in order to have a better understanding on what is remaining in the aqueous phase. Comparing the measured TOC with that calculated from the reaction intermediates detected allows visualizing if there are non-identified byproducts. Also, it would allow the reader to confirm rather quickly (at a glance) that acetic acid is the main product. This does not imply new experiments, only calculating the C content of each quantified intermediate and comparing it to the measured TOC.
5. Conclusions cannot include references, this should have already been discussed in the main text.
You should also take into account the dollowing for future works:
1. Carefully revise your language when answering the reviewers, they are trying to help you, they are not the enemy.
2. Not all toxicity tests are expensive. You can use Artemia salina to assess the toxicity of the influents and effluents. This is also known as sea monkey, they are available at pet stores and are really cheap. The inclussion of this information in the manuscript can greatly improve its soundness, specially since you are working at very high pollutant concentrations and simple water matrices.
3. Analysis of operating costs and comparison with other AOPs can improve the quality of the manuscript.
Author Response
Reviewer 3
Reviewer’s remark
- Units coherence. All along the manuscript, units are refered as g L-1. In Figure 3, SDS concentration is presented as mg/l. Uniformize the units format and revise throuhout the manuscript.
Authors’ response
The authors fulfilled the requirement in respect to Figures 3 and 5.
Reviewer’s remark
- Line 195, revise nomenclature for hydroxyl radical.
Authors’ response
The authors replaced ‘OH’ symbol with ‘hydridooxygen’ term as required by the reviewer #1 following recommendations of Koppenol, W. (2000). Names for inorganic radicals (IUPAC Recommendations 2000). Pure and Applied Chemistry 72(3): 437-446.
Reviewer’s remark
- Authors state that TOC removal is negligible with the following statement: "The latter circumstance explains also the fact that the TOC 214 analysis showed negligible changes within the experimental conditions. The refractory 215 character of NPX oxidation products was reported also by Rosal et al. [24]." In the article by Rosal et al., a 75% TOC reduction is reached. Hence, this does not explain at all your results.
Authors’ response
The 75% TOC reduction in the article by Rosal et al. was reached for carbamazepine, not naproxen (Fig. 4). Non-catalytic ozonation of naproxen resulted in no more than 45% of mineralization at the longest time of treatment/the largest energy expense. The very first data on TOC reduction for 10% shown in the article report changes in two minutes of treatment: gas flow 0.2 Nm3h-1 (3.3 L min-1) at gaseous ozone concentration of 40 g m-3 delivers 264 g m-3 in two minutes. Such amount of ozone at minimum energy expense of 10 kWh kg-1 with oxygen as a carrier gas requires 2.64 kWh m-3 of energy; with ozonized air this number is at least doubled, comprising 5.0-5.5 kWh m-3. The energy dose delivered to the PCD reactor working with air considered in the manuscript did not exceed 2.7 kWh m-3, being able to remove 5% of TOC at largest, i.e., within the observation accuracy limit. This makes the comparison impossible, as said in the manuscript: the authors simply are not able to see TOC changes at the energy dose applied, which supports the observation.
Reviewer’s remark
- The suggestion of performing a carbon balance is in order to have a better understanding on what is remaining in the aqueous phase. Comparing the measured TOC with that calculated from the reaction intermediates detected allows visualizing if there are non-identified byproducts. Also, it would allow the reader to confirm rather quickly (at a glance) that acetic acid is the main product. This does not imply new experiments, only calculating the C content of each quantified intermediate and comparing it to the measured TOC.
Authors’ response
The authors added Table 1 for TOC determined in samples by instrumental measurement and as a sum of theoretically expected at the energy doses of 0.9, 1.8 and 2.7 kWh m-3 for both starting pH 3.0 and 7.0.
Reviewer’s remark
- Conclusions cannot include references, this should have already been discussed in the main text.
Authors’ response
Corrected as required by the referee: the reference with relevant discussion was placed to the Results and Discussion text, the Conclusion part was shortened accordingly.
Reviewer’s remark
You should also take into account the dollowing for future works:
- Carefully revise your language when answering the reviewers, they are trying to help you, they are not the enemy.
- Not all toxicity tests are expensive. You can use Artemia salina to assess the toxicity of the influents and effluents. This is also known as sea monkey, they are available at pet stores and are really cheap. The inclussion of this information in the manuscript can greatly improve its soundness, specially since you are working at very high pollutant concentrations and simple water matrices.
- Analysis of operating costs and comparison with other AOPs can improve the quality of the manuscript.
Authors’ response
The authors have nothing against the reviewer’s advises per se and act accordingly within their possibilities, measuring toxicity with Vibrio fischeri tests and reporting energy efficiency of other methods, wherever necessary/possible.